# Shade, Aging and Spatial-Dependent Variation of Elastoplastic and Viscoelastic Characteristics in a Dental, Submicron Hybrid CAD/CAM Composite

**DOI:** 10.3390/ma16165654

**Published:** 2023-08-17

**Authors:** Nicoleta Ilie

**Affiliations:** Department of Conservative Dentistry and Periodontology, University Hospital, Ludwig-Maximilians-Universität in Munich, Goethestr. 70, D-80336 Munich, Germany; nilie@dent.med.uni-muenchen.de; Tel.: +49-89-44005-9412; Fax: +49-89-44005-9302

**Keywords:** CAD/CAM, resin-based composites, strength, Weibull statistics, dynamic-mechanical analysis, viscoelasticity

## Abstract

This article reports the elastoplastic and viscoelastic response of an industrially cured CAD/CAM resin-based composite (Brilliant Crios, Coltene) at different scales, spatial locations, aging conditions, and shading. Mechanical tests were performed at the macroscopic scale to investigate material strength, elastic modulus, fracture mechanisms and reliability. An instrumented indentation test (IIT) was performed at the microscopic level in a quasi-static mode to assess the elastic and plastic deformation upon indentation, either by mapping transverse areas of the CAD/CAM block or at randomly selected locations. A dynamic-mechanical analysis was then carried out, in which chewing-relevant frequencies were included (0.5 to 5 Hz). Characteristics measured at the nano- and micro-scale were more discriminative in identifying the impact of variables as those measured at macro scale. Anisotropy as a function of the spatial location was identified in all shades, with gradual variation in properties from the center of the block to peripheral locations. Depending on the scale of observation, differences in shade and translucency are very small or not statistically significant. The aging effect is classified as low, but measurable on all scales, with the same pattern of variation occurring in all shades. Aging affects plastic deformation more than elastic deformation and affects elastic deformation more than viscous deformation.

## 1. Introduction

Industrially, in block-cured CAD (Computer-Aided Design)/CAM (Computer Aided Manufacturing), resin-based composites (RBCs) belong to the newest category of dental restorative materials. Taking into account the inherent variation within a material category, both in situ light-cured and block-cured RBCs are based on similar chemical composition, with dimethacrylate monomers as the organic matrix, e.g., Bis-GMA (bisphenol A diglycidyl dimethacrylate), bis-EMA (ethoxylated bisphenol A dimethacrylate), UDMA (urethane dimethacrylate) and TEGDMA (triethylene glycol dimethacrylate), in which inorganic fillers are embedded [1].

Compared to the much more commonly used light-cured RBCs, CAD/CAM RBCs offer the perspective of absolute control over the polymerization quality and, thus, the production of larger and homogeneously cured restorations through subtractive manufacturing. These desiderates have been reached so far, as the controlled thermal curing and isostatic pressure in CAD/CAM RBCs resulted in an improvement in physical properties compared to the incrementally placed, light-cured RBCs [2] through increased monomer conversion and network density [3], potential improved filler-matrix interaction [4] and reduced air inclusions [2]. In addition, lower monomer release [2,5,6], toxicity [2], and plaque/biofilm formation have been reported compared to light-cured RBCs [7]. Related to glass-ceramic or ceramic CAD/CAM materials, CAD/CAM RBCs offer significant advantages in terms of machinability and intra-oral reparability [4] and have better damping capabilities over a wider frequency range [8]. In addition to the use in tooth reconstruction as inlays, onlays, crowns and veneers, simulated finite element models reveal that CAD-CAM RBCs are also suitable as an alternative material for implant-supported restorations in the posterior region [9].

In contrast to the advantages listed above, CAD/CAM RBCs are attributed to a certain anisotropy that has been quantified as a variation of up to 8.7% in micromechanical properties within different regions of a CAD/CAM block [10]. Because properties gradually vary from the central areas to the periphery of a block, either decreasing or increasing, the location of the block used to prepare a restoration can be an issue that requires attention. Whether the peripheral areas of a block show higher or lower values compared to the central area was not related to the chemical composition of the materials or the inorganic filler content but rather to the industrial curing process [10]. There is no clear reference to this aspect in the literature, but the limited available literature on CAD/CAM RBCs technology reports that many materials currently on the market use not only photo-initiators but also thermal catalysts to initiate polymerization [3]. The relationship with the observed variation in properties is, therefore, difficult to clarify since both light transmission and thermal gradients are to be expected. Other reported deficits in CAD/CAM RBCs are related to the effect of aging on various physical and mechanical properties, such as noticeable color changes that are still acceptable, small but visible changes in gloss, significantly increased surface roughness [11], or significantly reduced strength and reliability [12] are described. Contrasting with these results, a recent study attested CAD/CAM RBCs have higher resistance to cyclic fatigue compared to zirconia and lithium disilicate restorative materials and insignificant changes in surface roughness after exposure to erosive media [13].

When analyzing a material’s performance, the most common approach is to take a single shade and apply the results to shades across the brand without questioning a possible variation. However, certain variations are to be expected, especially with modern CAD/CAM RBCs, which are not only manufactured in different shades but also in different translucencies in order to meet increasing aesthetic demands.

The study aims, therefore, to verify whether shade and translucency variations within a submicron hybrid CAD/CAM RBC are reflected in its elastoplastic and viscoelastic behavior described on nano, micro and macro scale, in the spatial distribution of mechanical parameters and in the way the material ages. Particular attention was paid to the shades selected to accommodate the latest trend, the super translucent shade (ST), and to compare it with the already established highly translucent shades (HT). In addition, the selected material has already been intensively examined in previous publications and comparable test setups in a low translucent shade (LT), which not only serves as a literature comparison for the current study but also allows the results to be linked to other CAD/CAM RBCs.

The null hypotheses tested were that various shades of a CAD/CAM RBC behave similarly with respect to (a) strength, elastic modulus, reliability and fracture pattern; (b) elastoplastic behavior, (c) viscoelastic behavior, (d) spatial distribution of the properties and (e) accelerated thermal aging.

## 2. Materials and Methods

### 2.1. Specimen Preparation

A total of 120 parallelepiped specimens of ca. 2 mm × 2 mm × 18 mm were cut from CAD/CAM RBCs blocks belonging to three different shades of the same brand and composition (Table 1), while using 5 CAD/CAM blocks per shade. All four large surfaces of the samples were wet ground in an automatic grinding machine (EXAKT 400CS Micro Grinding System EXAKT Technologies Inc., Oklahoma City, OK, USA) with silicon carbide abrasive paper (P1200, P2500, P4000) to achieve the geometry described above with an accuracy of 0.1 mm. All specimens were stored in distilled water at 37 °C for 24 h. Half of the specimens for each shade (n = 20) were randomly selected from the specimen pool and subjected to three-point testing. The other half was additionally aged thermally (TA) for 10,000 cycles in distilled water between 5 and 55 °C at a dwell time of 30 s per temperature and a transfer time of 10 s between baths (Willytec, Dental Research Division, Munich, Germany).

Five additional CAD/CAM blocks of each shade were used to create approximately 2 mm-thick specimens by cutting the blocks parallel across the width to obtain a total of 45 specimens, which were ground similarly as described above. These samples were randomly divided into three groups per each shade (n = 5) and subjected to instrumented indentation testing (IIT). Therefore one-third of the samples were measured dry; the remainder were immersed in distilled water at 37 °C for 24 h. Half of these were randomly selected and tested, while the rest were thermally aged prior to IIT as described above.

### 2.2. Three-Point Flexural Test

A three-point bending test according to NIST No. 4877 was carried out with a distance of 12 mm between the supports [14] while using a universal testing machine (Z 2.5, Zwick/Roell, Ulm, Germany) at a crosshead speed of 0.5 mm/min in accordance to ISO 4049 [15]. The force in bending was measured as a function of beam deflection. The slope of this curve in the linear part allows the calculation of the flexural modulus.

### 2.3. Fractography Analysis

The fracture pattern and fracture origin were determined by means of a stereomicroscope (Stemi 508, Carl Zeiss AG, Oberkochen, Germany) and documented with a microscope extension camera (Axiocam 305 color, Carl Zeiss AG, Oberkochen, Germany). The origin of the fracture was identified as either a volume defect (sub-surface) or a surface defect located at either the edge or the corner of the specimens. In addition, the fracture mirror was defined with the mirror borders located at first noticeable roughness after the smooth mirror region, and the mirror radius was measured (ImageJ Version 1.53k, U.S. National Institutes of Health, Bethesda, MD, USA). The measured radius was placed in the direction of constant stress, parallel to the tensile side of the specimen, from the origin of fracture to the mirror borders. When the origin of the fracture could not be exactly identified, the diameter of the mirror was measured and halved. The mirror radius was used to determine the mirror constant A as described by the Orr equation [16]: $$\sigma \sqrt{R}=A$$
with σ = flexural strength, *R* = mirror radius, and *A* = mirror constant.

### 2.4. Instrumented Indentation Test (IIT)

(a) Quasi-Static Indentation test

The 45 specimens prepared and stored as described above were subjected to a quasi-static indentation test according to ISO 14577 [17] using an automated nano-indenter (Fischerscope^®^ HM2000, Ultraprecisao, Aveiro, Portugal) equipped with a Vickers diamond tip. Three measurements were randomly performed on each sample (n = 5). The indentation was performed with force control. Therefore, the test load increased from 0.4 mN to 1000 mN over 20 s. The maximal force was maintained for additional 5 s to then decrease over 20 s at a constant speed. Loading parameters have been selected according to ISO 14577 and previous publications [8,18]. Parameters characterizing the elastic and plastic material behavior were calculated from the variation of indentation force (F) and indentation depth (h) during a load-unload cycle. The slope of the tangent of the indentation depth curve at the maximum force was used to calculate the indentation modulus E_IT_. Hardness was calculated by accounting for the impression created during the indentation while differentiating between the projected indenter contact area (A_c_) and the surface area of the indentation under the applied test load (A_s_). A_c_ was determined from the force–indentation depth curve according to ISO 14577 [17]. The resistance to plastic deformation was then designated by the indentation hardness (H_IT_ = F_max_/A_c_) and the Vickers hardness (HV = 0.0945 × H_IT_). In addition, the universal hardness (or Martens hardness = F/A_s_(h)) was calculated by dividing the test load by A_s_ and characterizes both plastic and elastic deformation. Furthermore, the creep was calculated from changes in indentation depth during the 5 s of maintaining maximal indentation force during the indentation process described above. The work of indentation (W_tot_) was calculated as the integral of the force with the depth (=∫Fdh) and divided into its plastic (W_plast_ = mechanical work consumed as plastic deformation) and elastic (W_elast_ = mechanical work consumed as elastic reverse deformation) components. These parameters enable determining a prerequisite variable for the further DMA test (W_elast_/W_tot_ = µ_IT_).

In addition, the distribution of the parameters described above was assessed by mapping in 500 µm increments a randomly selected transverse region from a CAD/CAM block of each shade, amounting to 572 indentations per sample. The surface was prepared as described above.

(b) Dynamic mechanical analysis (DMA)

The 45 specimens previously subjected to the quasi-static indentation test were used to assess the DMA material parameters while superimposing onto a quasi-static force of 1000 mN and a low-magnitude oscillating force at 10 different frequencies in the range 0.5–5 Hz, to include the chewing frequency in humans (0.94 Hz to 2.17 Hz [19]). To keep deformation within the linear viscoelastic regime, the oscillation amplitude was set at five nm [8,18].

Fifteen individual measurements were performed per shade and aging conditions while selecting their position randomly. Ten measurements were performed at each indentation and at each of the 10 used frequencies. The sinusoidal response signal was used to calculate the storage (E′), the loss moduli (E″) and the loss factor (tan δ).

### 2.5. Statistical Analyses

The normality of the acquired data was confirmed using the Shapiro–Wilk procedure. The effect strength of the parameters shade, aging condition and frequency, as well as their interaction terms, was assessed by means of a multivariate analysis (general linear model). The partial eta-squared statistic reported the practical significance of each term based on the ratio of the variation attributed to the effect. Larger values of partial eta-squared (η_P_^2^) indicate a greater amount of variation accounted for by the model. The results were further compared using multiple-way analysis of variance (ANOVA) and Tukey honestly significant difference (HSD) post-hoc test (α = 0.05 (SPSS Inc. Version 29.0, Chicago, IL, USA). Flexural strength data were additionally subjected to Weibull analysis to determine material reliability [20].

## 3. Results

### 3.1. Three-Point Flexural Test and Fractography Analysis

The parameters measured in the three-point flexural test are summarized in Table 2. A multifactorial ANOVA evidenced that the parameter shade had no effect on FS (*p* = 0.52) and E (*p* = 0.065), but the effect of aging was significant on both parameters (*p* < 0.001) and stronger on FS (η_P_^2^ = 0.332) than on E (η_P_^2^ = 0.127). Weibull analysis revealed slightly lower reliability for BL LT, while the Weibull modulus m was similar for BL ST and C2 LT (confidence intervals overlapped). Aging significantly reduces reliability for BL ST only.

#### Fractography Analysis

The fracture mode was identified as initiated from a flaw on the tensile side of the specimen, either located at the surface (edge and corner) or sub-surface, with mirror, mist, and Hackle lines forming as the crack propagated (Figure 1). The distribution of the fracture modes is summarized in Figure 2 and reveals that failures due to surface defects were the most common failure mode (70%) both after 24 h storage and after aging, with volume defects (sub-surface mode) accounting for 30%. A difference between the aging modes was registered only within surface defect types, with corner defects accounting for 23.3% and edge defects at 46.7% after 24 h of storage, and 13.3% and 56.7%, respectively, after thermal aging (TA).

### 3.2. Instrumented Indentation Test (IIT): Quasi-Static Approach

Within each shade, one-way ANOVA identified a significant decrease in HV, HM and E_IT_ in the sequence dry > 24 h > TA. In contrast, an increase in the same order was measured for the parameters W_e_, W_tot_ and Cr (Table 3).

A multifactorial analysis revealed a significant (*p* < 0.001, Table 4) effect of aging and shade (except for W_e_, *p* = 0.361) on the measured parameters, while the effect of aging was higher.

The spatial distribution of the quasi-static IIT parameters is illustrated exemplary for C2 LT and the parameters HM, HV, E and Creep in Figure 3. In all shades, slightly higher HV, HM, E_IT_ and lower Cr, W_e_ and W_tot_ were measured in the central compared to the marginal region. The inhomogeneity of the parameters is lower 8% in all shades.

### 3.3. Instrumented Indentation Test (IIT): Dynamic Mechanical Analysis (DMA)

A multifactorial analysis revealed a significant (*p* < 0.001, Table 5) influence of aging, shade and frequency on all parameters, with the influence of aging being the highest, followed by frequency and with a very low impact, the shade. Aging affected stronger H_IT_ and E′, followed by tan δ, while the lowest influence was on E″. The binary combined effect of aging × shade was consistently lower compared to the effect of the individual parameters but still significant. All other combined effects have no impact on H_IT_ and very little impact on the other measured parameters.

Figure 4 and Figure 5 indicate a similar variation pattern of the measured parameters with frequency for all shades and aging conditions. Shade comparison reveals low or insignificant differences, with H_IT_ curves increasing and E′, E″, and loss factor decreasing exponentially with increasing frequency until a plateau at 1.4 Hz.

The aging effect is exemplified for shade BL ST and was more pronounced for the H_IT_ parameter, with the highest values being obtained with dry samples and the lowest with thermally aged samples. The effect is reproduced congruently in E′, becoming more prominent with increasing frequency. E″ and loss factor followed the described trend, while the dry state and 24 h water immersion were often comparable.

## 4. Discussion

The initial concerns about the clinical necessity and suitability of CAD/CAM RBCs have already been addressed [21,22], and their functionality in the oral environment has been confirmed. The next development step was to expand the color and translucency portfolio in such a way that the CAD/CAM RBCs are in no way inferior to their light-curing counterparts in terms of aesthetics. This last aspect is clinically important as finding the right shade that perfectly matches a tooth is fairly easy, but noticeable aesthetic problems are mostly related to the mismatch in translucency between the restorative material and the tooth. Therefore, the present study characterized three different shades with similar chemical composition, taking the latest development, a super translucent shade (BL), to compare it with two low translucent versions in a light (BL) and a darker (C2) shade. Since all three shades are based on the same composition, it is believed that the super translucency was created by matching the refractive index of the filler and matrix to reduce scattering at the filler/matrix interface [23], and by adjusting the size of the fillers to optimize the dimensions of this interface. Furthermore, curing in CAD/CAM RBC blocks is not exclusively related to photo-initiation or blue light [3]; therefore an oversupply of intensely yellow camphorquinone, as is inevitable with light-cured RBCs, is no longer an impediment. Lower translucencies in RBCs are then usually achieved simply by adding a small amount of opacifying agents, mostly metal oxides such as titanium oxide (TiO_2_), aluminum oxide (Al_2_O_3_) or zirconium oxide (ZrO_2_) with supposed negligible contribution to the mechanical properties. In fact, this was confirmed by the result of the three-point bending test, as statistically similar mechanical parameters were observed in all shades. In addition, the data measured in the present study for low and super translucent shades (Table 1) are consistent with literature data for the same material but with a high translucent shade (HT A3; FS = 239.0 ± 22.9 MPa, dry, and 201.9 ± 11.9 MPa, after thermal aging) [12]. This supports the conclusion that the FS is independent of shade within the given material composition. The direct comparison with literature data is possible here because similar setups, sample geometries and aging conditions have been used.

The three-point bending test specimens were additionally subjected to a qualitative and quantitative fractographical analysis [24] based on the proven suitability of highly filled RBCs for applying the principles of brittle fracture mechanics [25,26]. In fact, the brittle fracture characteristics are clearly cognoscible in Figure 1, where the fracture originates from a critical flow, located either at the surface or in the volume of the specimen, and then propagates to form a smooth radial region, the fracture mirror [27]. As the crack velocity further increases, the crack tip deviates from the main plane, giving rise to secondary cracks that are unable to propagate due to the decreasing energy [28] and therefore produce a characteristic, stippled pattern (mist) [29]. The mist area is followed by an area of larger radial ridges (Hackle lines) leading to macroscopic crack branches [30]. The quantitative evaluation of the fracture mirror enables the calculation of a parameter, the mirror constant, which can be related to the fracture toughness K_Ic_ [31,32] and can then be used clinically on more complex geometries [33,34]. The calculated mirror constant was statistically the same for all shades (overlapping of the 95% CI in Table 1) and can be given as a mean value of the tested material of 2.58 MPa√m after 24 h storage in distilled water and 2.49 MPa√m after aging. These values agree well with values measured with an indirect RBC (2.6 MPa√m) [25] and are in the range of light-cured RBCs (2.23 to 3.39 MPa√m [26]), and ceramics (0.97 to 6.6 MPa√m [24]).

The variation of mechanical parameters within a CAD/CAM block was evaluated by densely mapping a cross-section of a block in each shade. A gradual variation in properties from the center of the block to peripheral locations has been observed in all shades, with slightly higher HV, HM, E_IT_ and corresponding lower Cr, W_e_ and W_tot_ identified in the central regions compared to the peripheral region. These results are consistent with data previously measured in other CAD/CAM RBCs like Lava Ultimate (3M), Grandio Blocs (VOCO), and Tetric CAD (Ivoclar Vivadent) [10], possibly as a result of a similar curing technology in terms of exposure to light and/or pressure during polymerization. In contrast to this behavior, improved curing in the peripheral areas compared to the center has so far only been reported for one CAD/CAM RBC block (Shofu Blocks, Shofu) [10]. As detailed information on the curing technology of the blocks are missing, these effects are attributed to light attenuation during polymerization, temperature and pressure gradients, and most likely to accumulated residual stress [35].

The indicated elastoplastic parameters at the microscopic scale reflect local material variations and not the mechanical properties of the individual components, i.e., the filler or the organic matrix. The proof of this statement is provided by the indentation depth, which was between 8 and 9 µm. The indenter used was a Vickers pyramid, the indentation of which in the material can be estimated as a rhombus with diagonals approximately seven times larger than the indentation depth measured at maximum loading. Ultimately, this means a diagonal of ca. 56 µm to 63 µm and is, therefore, consistently smaller than the filler sizes in CAD/CAM RBCs [10]. This endorsed identifying even small variations of the material within different locations in a CAD/CAM block or sample.

In fact, based on the evaluated micromechanical parameters, a significant shade effect can be identified, which was consistently less than the aging effect (Table 4, lower partial eta-squared values). To put this difference in perspective, it is necessary to consider that the aging effect compared to the dry material does reduce the measured parameters but to a value of less than 7%. Since the statistically calculated effect of the shade in connection with the aging effect is consistently lower, the difference within shades can be defined as measurable at the micro-scale versus macro-scale (three-point bending test) but very small. Similar considerations apply to the dynamic-mechanical parameters (nano-scale).

When evaluating the performance of a material, it must be considered that dental restorations are subjected to thermal and mechanical stresses after placement in the oral cavity, in addition to degradation induced by hydrolysis of the ester groups in methacrylates and catalyzed by enzymes [36,37,38], or acids and bases [39] present in the dietary components. The hydrolytic biodegradation is not limited to the polymer matrix but also includes the adhesion of the inorganic filler in the methacrylate matrix [40] provided by amphiphilic silanes coupled to the organic matrix via their methacrylate functional group and to the inorganic fillers via their alkoxy functional group. Both functional groups, ester and alkoxy, tend to degrade over time in a humid environment, accelerating RBC degradation [41]. A laboratory simulation of aging, including water uptake and concomitantly stress induction from cyclic thermal changes, consists of subjecting the material to an artificial aging protocol of cold (5 °C) and warm (55 °C) water immersion alternations. The aging effect in light-cured RBCs is usually related to the data measured after 24 h of storage in distilled water, a protocol that was transposed in the present study to the CAD/CAM RBCs. In contrast to light-cured RBCs, which undergo post-polymerization within the 24 h after exposure to light associated with improvement in mechanical properties, storage in water for 24 h of a priori cured CAD/CAM RBCs that have already experienced physical aging, appears to slightly decrease these parameters. Water uptake during this initial short period is already indicative of the occurrence of plasticization and polymer density reduction since slight changes in properties are perceived, albeit often bordering on the level of significance. After thermal fatigue and prolonged exposure to water, there is further, slight degradation of the materials. It is noteworthy that the micro-mechanical properties, which are aligned with plastic features of the deformation (HV, Cr, W_tot_), are more affected by degradation than the elastic components (E, W_e_). This trend is congruent with the DMA data, emphasizing a greater impact of aging on hardness (H_IT_) vs. elastic modulus (E) and confirming the aging behavior of many light-cured RBCs [18]. Likewise, it can be inferred that the effect of aging has a greater impact on the elastic compared to the viscous material response. The influence of aging on viscous behavior is also small but is clearly reflected in the increased energy dissipation potential (tan delta and loss modulus E″) [42] as a cumulative effect of polymer plasticization and friction at interphase boundaries [43].

A large number of parameters have been analyzed in order to filter out differences in the elastoplastic and visco-elastic mechanical behavior of different shades of material. However, the characterization is limited by the lack of clinical data that would allow for the validation of the present results.

## 5. Conclusions

The differences between shades and translucencies within a given material composition can be considered non-existent or very small depending on the scale of observation. In contrast, aging, even when the effect was small, could be quantified at the macro, micro, and nano-scales and followed the same pattern of variation in all shades. Aging affects plastic deformation more than elastic deformation and affects elastic deformation more than viscous deformation. Anisotropy of the measured parameters was observed within a composite block and in all shades.

## Figures and Tables

**Figure 1 materials-16-05654-f001:**
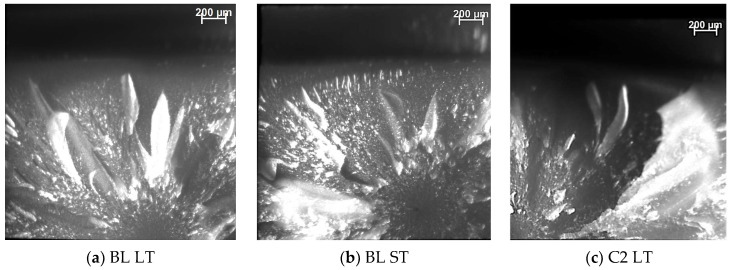
Fracture modes exemplified on (**a**) BL LT (Flexural strength, FS = 224.9 MPa; fracture initiated by a defect located at the edge); (**b**) BL ST (FS = 242.6 MPa; fracture initiated by a sub-surface defect) and (**c**) C2 LT (FS = 236.4 MPa, fracture initiated by a defect located at the corner).

**Figure 2 materials-16-05654-f002:**
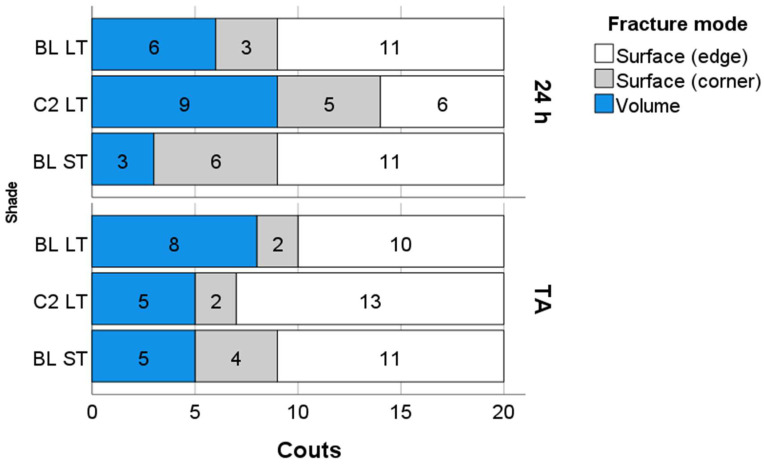
Fracture mode distribution among analyzed shades.

**Figure 3 materials-16-05654-f003:**
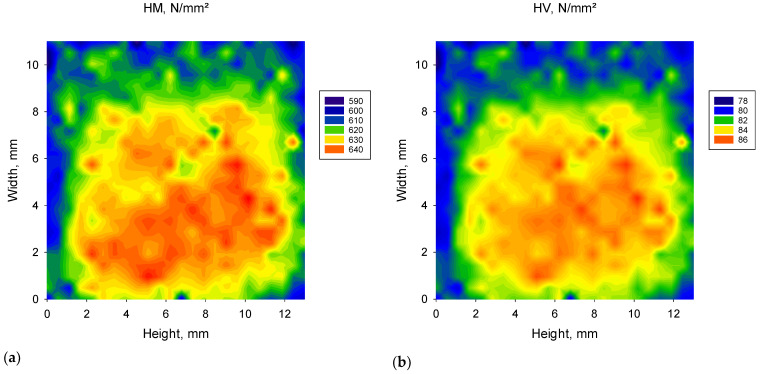
Spatial distribution of the IIT parameters measured in quasi-static mode (**a**) Martens Hardness—HM; (**b**) Vickers hardness—HV; (**c**) Indentation modulus—E_IT_; and (**d**) Creep, exemplified on a transverse section through a CAD/CAM block of shade C2 LT.

**Figure 4 materials-16-05654-f004:**
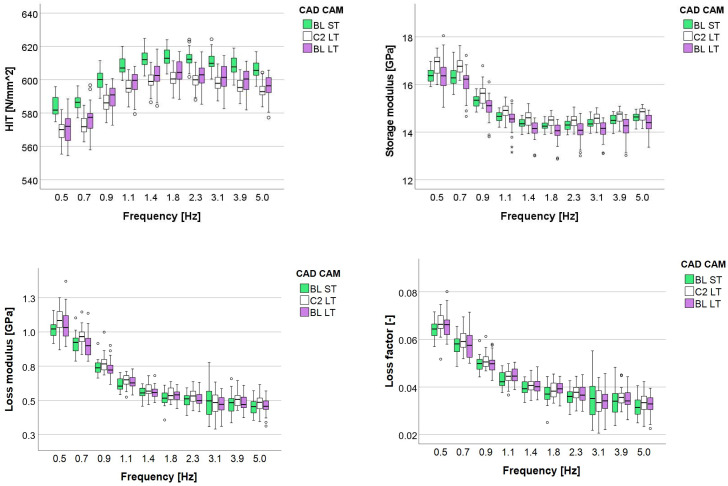
Dynamic mechanical analysis: Variation of the indentation hardness H_IT_, storage modulus, loss modulus, and loss factor over the frequency range 0.5–5 Hz for the analyzed shades.

**Figure 5 materials-16-05654-f005:**
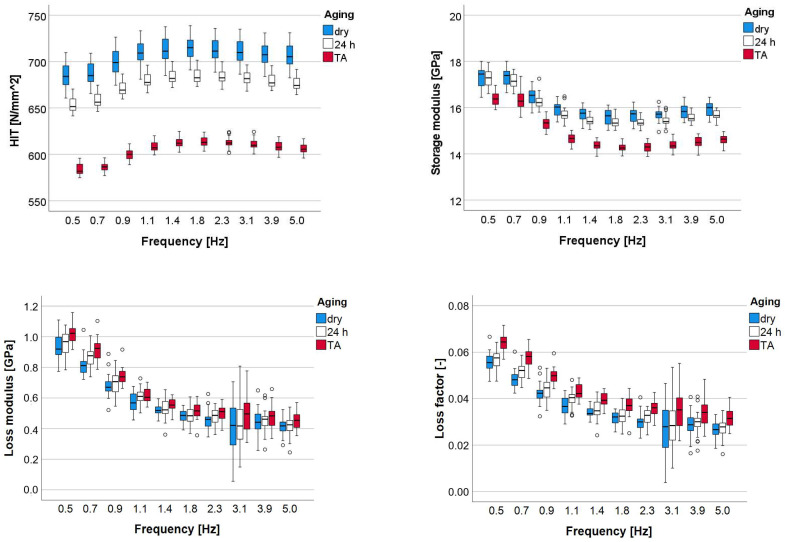
Dynamic mechanical analysis: Variation of the indentation hardness H_IT_, storage modulus, loss modulus, and loss factor over the frequency range 0.5–5 Hz for the analyzed aging conditions exemplified for shade BL ST.

**Table 1 materials-16-05654-t001:** CAD/CAM resin-based composite Brilliant Crios: shade, translucency, LOT number and composition are as indicated by the manufacturer (Coltene, Altstätten, Switzerland).

Shade (Translucency)	LOT	Composition		wt%	vol%
		Matrix	Filler		
BL ST (super translucent)	K82874	BisGMA UDMA TEGDMA	BaO-Al_2_O_3_-SiO_2_ SiO_2_	70.7	51.5
C2 LT (low translucent)	K76951				
BL LT (low translucent)	K59845				

Abbreviations: BisGMA—bisphenol A glycol dimethacrylate; UDMA—urethane dimethacrylate; TEGDMA—triethylene glycol dimethacrylate; BaO-Al_2_O_3_-SiO_2_—barium aluminosilicate glass; SiO_2_—silicon oxide (silica), wt%—weight percent; vol%—volume percent, BL—Brilliant Crios; ST—super translucent; LT—low translucent.

**Table 2 materials-16-05654-t002:** Three-point bending test; Flexural strength—FS, mirror constant A (95% confidence interval CI), Weibull parameters (95% CI and R-squared (R^2^) values), flexural modulus E (mean and standard deviation—SD). Values denoted by the same superscript are statistically similar.

Shade	Aging	FS, MPa		A, MPa√m	Weibull Parameters		E, GPa	
		Mean	SD		m	R^2^	Mean	SD
BL ST	24 h	232.2 ^a^	15.4	2.46–2.61	16.5–19.6	0.97	7.5 ^a^	0.6
	TA	203.1 ^b^	17.1	2.26–2.41	13.4–15.3	0.98	7.2 ^b^	0.4
C2 LT	24 h	232.5 ^a^	13.2	2.43–2.57	18.6–22.7	0.96	7.5 ^a^	0.4
	TA	210.4 ^b^	14.5	2.43–2.60	14.6–19.3	0.92	6.6 ^b^	1.0
BL LT	24 h	229.3 ^a^	25.0	2.44–2.58	8.1–11.2	0.90	7.3 ^a^	0.6
	TA	205.0 ^b^	21.5	2.28–2.45	9.9–12.1	0.96	7.2 ^b^	0.4

**Table 3 materials-16-05654-t003:** Quasi-static parameters as a function of shade and aging conditions (TA—thermal aging): HM—Martens hardness, HV—Vickers hardness; E_IT_—indentation modulus; μ_IT_—W_e_/W_t_; W_e_—elastic indentation work; W_tot_—total indentation work; Cr—Creep; SD—standard deviation.

Shade	Aging		HM N/mm^2^	HV N/mm^2^	E_IT_ GPa	μ_IT_ %	We µJ	Wtot µJ	Cr %
BL ST	dry	Mean	572.4	87.2	12.4	48.3	1.3	2.7	4.1
		SD	3.5	0.6	0.1	0.3	0.01	0.01	0.02
	24 h	Mean	541.0	82.6	11.6	48.9	1.4	2.8	4.2
		SD	2.7	0.6	0.0	0.2	0.00	0.01	0.04
	TA	Mean	533.5	80.2	11.7	47.8	1.3	2.8	4.3
		SD	5.7	0.8	0.2	0.5	0.0	0.0	0.0
C2 LT	dry	Mean	554.0	83.6	12.1	47.9	1.3	2.8	4.3
		SD	6.9	1.2	0.1	0.3	0.01	0.02	0.05
	24 h	Mean	545.7	81.4	12.1	47.7	1.3	2.8	4.4
		SD	7.0	1.2	0.1	0.2	0.01	0.01	0.04
	TA	Mean	514.1	76.4	11.3	48.0	1.4	2.9	4.6
		SD	7.8	1.0	0.2	0.5	0.02	0.02	0.02
BL LT	dry	Mean	562.6	85.8	12.2	48.6	1.3	2.7	4.1
		SD	5.8	1.1	0.1	0.3	0.01	0.01	0.03
	24 h	Mean	535.9	80.0	11.8	48.0	1.3	2.8	4.4
		SD	3.5	0.5	0.1	0.2	0.0	0.02	0.03
	TA	Mean	514.5	76.7	11.3	47.4	1.4	2.9	4.5
		SD	7.1	1.0	0.2	0.4	0.02	0.01	0.03

**Table 4 materials-16-05654-t004:** Quasi-static IIT: effect strength of the parameters of influence. Partial eta-squared values η_P_^2^ are indicated when the effect was significant (*p* < 0.001); n.s. is reported if the effect was not significant.

Parameter	HM N/mm^2^	HV N/mm^2^	E_IT_ GPa	μ_IT_ %	We µJ	Wtot µJ	Cr %
Aging	0.905	0.926	0.829	0.332	0.593	0.926	0.931
Shade	0.471	0.674	0.112	0.292	n.s.	0.375	0.883
Aging × Shade	0.431	0.380	0.531	0.452	0.460	0.558	0.739

**Table 5 materials-16-05654-t005:** DMA IIT: effect strength. Partial eta-squared values η_P_^2^ are indicated when the effect was significant (*p* < 0.001); n.s. is reported if the effect was not significant.

Parameter	H_IT_	E′	E″	tan δ
Aging	0.962	0.799	0.127	0.349
Shade	0.199	0.13	0.029	0.011
Frequency	0.575	0.839	0.880	0.841
Aging × Shade	0.163	0.117	0.019	0.016
Frequency × Shade	n.s.	0.009	0.007	0.003
Aging × Frequency	n.s.	0.058	0.037	0.031

## Data Availability

Data are available on request.

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
