# Peer review of "Shade, Aging and Spatial-Dependent Variation of Elastoplastic and Viscoelastic Characteristics in a Dental, Submicron Hybrid CAD/CAM Composite"

_materials, 2023, doi:10.3390/ma16165654_

Round 1

Reviewer 1 Report

The new version of this work needs to address a few key areas before submission.

1. Although there is nothing wrong with the abstract itself, the writers should elaborate further on the data and commentary that is offered there. The abstract may benefit from the addition of some more numerical results.

2. likewise, the CAD/CAM resin-based composite material needs to be identified when it is acknowledged for the very first time in the abstract.

3. The author would be wise to offer a straightforward explanation for the process of preparing the documents, paying particular attention to the information provided in the table. I believe that the reader will find it to be quite confusing.

4. Author much appreciates the clarity brought by the illustration in Figure 3. Make an effort to fit items a–d into a single image of consistent resolution.

5. The effects of shadow and time on the quasi-static characteristics are shown in Table 3. Thus, the standard deviation parameter should be displayed for all measurements.

6. The conclusion part is too short and does not contain any obtained results and / or recommendations. Hence, it is recommended that the conclusion section be written in paragraph form rather than using bullet points. The most important findings must also be included in the section that is devoted to drawing conclusions.

7. There are just a few typos and grammatical errors discovered throughout the text; however, the writers are respectfully requested to verify the entire corrected text for any remaining errors.

There are just a few typos and grammatical errors discovered throughout the text.

Author Response

All comments to the corresponding author have been addressed independently below. The authors’ rebuttal is always in BLUE and where changes have been added to the revised manuscript in light of the reviewer comments these are presented in RED.

The author would firstly like to thank the reviewers’ for taking the time to read and critically appraise the manuscript and secondly to thank the reviewers’ for their positive constructive comments in improving the work.

Reviewer 1 comments:

Comments and Suggestions for Authors

The new version of this work needs to address a few key areas before submission.

  1. Although there is nothing wrong with the abstract itself, the writers should elaborate further on the data and commentary that is offered there. The abstract may benefit from the addition of some more numerical results.

Author’s response: Thanks for the observation that I can understand. The abstract was written following the guidelines for Materials, which requires more a narrative summary of the work to catch the reader than clear statistical data, as we are used to in dental research papers. The amount of data in this study is very large so concrete values ​​are difficult to pick out. Please also take into account the severe limitation of the number of words for the abstract. It was more important to contour the main trends.

  1. likewise, the CAD/CAM resin-based composite material needs to be identified when it is acknowledged for the very first time in the abstract.

Author’s response: Thank you for the observation - I have now noted the name of the material as recommended.

  1. The author would be wise to offer a straightforward explanation for the process of preparing the documents, paying particular attention to the information provided in the table. I believe that the reader will find it to be quite confusing.

Author’s response: I have thoroughly reviewed the paper from the recommended perspective.  There is indeed a lot of data, but I have presented it very systematically in tables and graphs so that it can be well followed and understood.

  1. Author much appreciates the clarity brought by the illustration in Figure 3. Make an effort to fit items a–d into a single image of consistent resolution.

Author’s response: Thank you for this advice. The 4 graphs in Fig. 3 were now combined as recommended, resulting in a clear and aesthetic presentation.

  1. The effects of shadow and time on the quasi-static characteristics are shown in Table 3. Thus, the standard deviation parameter should be displayed for all measurements.

Author’s response: The standard deviation is presented in Table 3 for all parameters in the second line. For each material and condition there are two lines – the upper line in the mean, the lower line SDI apologize for forgetting to explain the abbreviation SD. Thank you for this important observation.

  1. The conclusion part is too short and does not contain any obtained results and / or recommendations. Hence, it is recommended that the conclusion section be written in paragraph form rather than using bullet points. The most important findings must also be included in the section that is devoted to drawing conclusions.

Author’s response: I have to strongly disagree with this comment - the conclusions are rather extensive and are closely related to the results and the hypothesis of the study. They emphasize the differences between the shades and translucencies as well as the influence of aging and the measurement scale – nano, micro, and macro. The anisotropy within the CAD/CAM blocks is mentioned as well as the way in which the elastoplastic and viscoelastic behavior is affected.

  1. There are just a few typos and grammatical errors discovered throughout the text; however, the writers are respectfully requested to verify the entire corrected text for any remaining errors.

 Author’s response: I would like to thank you for the careful review of the document in terms of content and language. I have checked the entire text thoroughly for errors of any kind. Many thanks for your effort.

Reviewer 2 Report

Review of “Shade, aging and spatial dependent variation of elastoplastic and viscoelastic characteristics in a dental, submicron hybrid CAD/CAM composite” by Nicoleta Ilie

The article presents the mechanical characterization of a CAD/CAM resin-based composite for dental application. Different kinds of methods (including 3-point flexural, indentation and DMA tests) are employed to examine bulk and superficial mechanical features at different relevant scales along with the regional dependency. Fractography analysis is presented by means of stereomicroscopy. Accelerated thermal aging was carried out and its effects on the elastoplastic and viscoelastic features are reported and discussed.

The paper deals with an interesting subject and is in overall well written. One could argue for long about the management of the study and make constructive suggestions that would entail further work. Nonetheless, the manuscript is mature in content and is of quality. The methodology is comprehensively presented and the statistical analysis seems well executed. The paper deserves in my opinion to be part of the state-of-the-art. Therefore, it is recommended for publication, but the following comments and remarks should be addressed:

1) The novelty compared to other papers published previously by the author should be better highlighted especially in the introduction section. The author published several papers on the characterization of the same material and the present contribution appears as a concatenation of the previous papers. It is expected that the author clearly discusses the position of this work versus the other papers.

2) The paper would be much stronger if some examples of 3-point flexural and indentation plots are presented in terms of load evolution.

3) It would be also great if the authors may add a picture of the different techniques of mechanical characterization as well as samples used in this study.

4) The choice of loading parameters must be explained.

5) You can introduce a limitation section in a relevant place (may be before the conclusion) in order to highlight the main drawbacks of the study.

6) Are there further outlooks based on the presented paper?

7) The different acronyms should be better explained, especially BL ST, C2 LT and BL LT.

8) The paper should be carefully checked. Small misprints are present. For example, the word “therefor” in page 2 should be “therefore”.

The paper should be carefully checked. Small misprints are present. For example, the word “therefor” in page 2 should be “therefore”.

Author Response

All comments to the corresponding author have been addressed independently below. The authors’ rebuttal is always in BLUE and where changes have been added to the revised manuscript in light of the reviewer comments these are presented in RED.

The author would firstly like to thank the reviewers’ for taking the time to read and critically appraise the manuscript and secondly to thank the reviewers’ for their positive constructive comments in improving the work.

Comments and Suggestions for Authors

Reviewer 2 comments:

Review of “Shade, aging and spatial dependent variation of elastoplastic and viscoelastic characteristics in a dental, submicron hybrid CAD/CAM composite” by Nicoleta Ilie

The article presents the mechanical characterization of a CAD/CAM resin-based composite for dental application. Different kinds of methods (including 3-point flexural, indentation and DMA tests) are employed to examine bulk and superficial mechanical features at different relevant scales along with the regional dependency. Fractography analysis is presented by means of stereomicroscopy. Accelerated thermal aging was carried out and its effects on the elastoplastic and viscoelastic features are reported and discussed.

The paper deals with an interesting subject and is in overall well written. One could argue for long about the management of the study and make constructive suggestions that would entail further work. Nonetheless, the manuscript is mature in content and is of quality. The methodology is comprehensively presented and the statistical analysis seems well executed. The paper deserves in my opinion to be part of the state-of-the-art. Therefore, it is recommended for publication, but the following comments and remarks should be addressed:

Author’s response:  Thank you for appreciating the work presented.

1) The novelty compared to other papers published previously by the author should be better highlighted especially in the introduction section. The author published several papers on the characterization of the same material and the present contribution appears as a concatenation of the previous papers. It is expected that the author clearly discusses the position of this work versus the other papers.

Author’s response:  Indeed, we published several articles on the CAD/CAM RBCs and the material analyzed in the present study has been analyzed in a different shade in one of our previous publications, which also served as a reference for the presented data. The present work aimed to identify if there are differences between the shades within a single brad, particularly with regard to the latest introduced shades, the “super-translucent”. I have emphasized this aspect as suggested and clarified and explained the link to our previous publications. The changes are noted in the penultimate paragraph of the introduction. Thank you for the advice.

2) The paper would be much stronger if some examples of 3-point flexural and indentation plots are presented in terms of load evolution.

Author’s response:  Thank you for the advice - this would be very easy to do, however as the differences between the shades are small, these graphs would not be very exciting in my view, as a stress-strain diagram (3-point bending) or a force-depth diagram (micro indentation) is a fairly common recording. In addition to this aspect, it must be noted that the number of graphs in the present paper is very large, although I have made a lot of effort to present the very large amount of data as compressed as possible and to group them in tables.

3) It would be also great if the authors may add a picture of the different techniques of mechanical characterization as well as samples used in this study.

Author’s response:  With similar arguments to the above, slabs (2mm x 2mm x 18mm) and slices cut out from CAD/CAM blocks can be very easily visualized by the reader. Because the tests are based on standards that are widely used for material characterization, the techniques/pictures are described in detail in the cited standards.

4) The choice of loading parameters must be explained.

Author’s response:  Thank you for the advice. I have now highlighted the loading parameters for each method, which are related to the cited standards and the relevant literature.

5) You can introduce a limitation section in a relevant place (may be before the conclusion) in order to highlight the main drawbacks of the study.

Author’s response: Thank you for this remark. A limitation section was introduced at the end of the discussion chapter.

6) Are there further outlooks based on the presented paper?

Author’s response:  The data and behavioral tendencies are presented and discussed in detail in the work and then summarized in the conclusions, so that in my opinion an extension is superfluous.

7) The different acronyms should be better explained, especially BL ST, C2 LT and BL LT.

Author’s response:  Thanks for this advice. The abbreviations have been explained at the end of Table 1. They were originally noted in the Table. Please note that they are part of the material name given by the manufacturer, while just BC is an acronym I have introduced for Brilliant Crios to make writing easier and avoid advertising a brand name.

8) The paper should be carefully checked. Small misprints are present. For example, the word “therefor” in page 2 should be “therefore”.

Author’s response: Thank you for reading the article carefully, also in terms of language. It was checked again and searched for errors.  

Comments on the Quality of English Language

The paper should be carefully checked. Small misprints are present. For example, the word “therefor” in page 2 should be “therefore”.

Author’s response: Thank you for pointing out the typo.

Round 2

Reviewer 2 Report

I thank the author for the responses to my previous comments. Most of required alterations have been carried out.

Therefore, I recommend the paper for publication.